# Evaluation of the Possibility of Using Hydroponic Cultivations for the Removal of Pharmaceuticals and Endocrine Disrupting Compounds in Municipal Sewage Treatment Plants

**DOI:** 10.3390/molecules25010162

**Published:** 2019-12-31

**Authors:** Daniel Wolecki, Magda Caban, Magdalena Pazda, Piotr Stepnowski, Jolanta Kumirska

**Affiliations:** Department of Environmental Analysis, Faculty of Chemistry, University of Gdansk, Wita Stwosza 63, 80-308 Gdańsk, Poland; daniel.wolecki@phdstud.ug.edu.pl (D.W.); magda.caban@ug.edu.pl (M.C.); magdalena.pazda@phdstud.ug.edu.pl (M.P.); piotr.stepnowski@ug.edu.pl (P.S.)

**Keywords:** pharmaceuticals, endocrine disrupting compounds, hydroponic cultivation, determining target pollutants in plant materials, municipal wastewater treatment plants

## Abstract

The problem of the presence of pharmaceuticals and endocrine disrupting compounds (EDCs) in the environment is closely related to municipal wastewater and in consequence to municipal wastewater treatment plants (MWWTPs) because wastewater is the main way in which these compounds are transferred to the ecosystem. For this reason, the development of cheap, simple but very effective techniques for the removal of such residues from wastewater is very important. In this study, the analysis of the potential of using three new plants: *Cyperus papyrus* (Papyrus), *Lysimachia nemorum* (Yellow pimpernel), and *Euonymus europaeus* (European spindle) by hydroponic cultivation for the removal of 15 selected pharmaceuticals and endocrine disrupting compounds (EDCs) in an MWWTP is presented. In order to obtain the most reliable data, this study was performed using real WWTP conditions and with the determination of the selected analytes in untreated sewage, treated sewage, and in plant materials. For determining the target compounds in plant materials, an Accelerated Solvent Extraction (ASE)-Solid-Phase Extraction (SPE)-GC-MS(SIM) method was developed and validated. The obtained data proved that the elimination efficiency of the investigated substances from wastewater was in the range of 35.8% for diflunisal to above 99.9% for paracetamol, terbutaline, and flurbiprofen. *Lysimachia nemorum* was the most effective for the uptake of target compounds among the tested plant species. Thus, the application of constructed wetlands for supporting conventional MWWTPs allowed a significant increase in their removal from the wastewater stream.

## 1. Introduction

Pharmaceuticals are used in large quantities around the world. In many cases, they are used not only to prevent disease or for treatment in humans, but also in animals [1,2]. Some of them, for example, natural and synthetic estrogens, are classified also to the group of endocrine disrupting compounds (EDCs). Over the last two decades, the presence of pharmaceuticals and EDCs in the aquatic environment has been confirmed many times [3,4,5,6]. They have been found in treated wastewater [7], sewage sludge [8], marine waters [9], and in living organisms [10]. They are extensively introduced into the environment via wastewater and in consequence by wastewater treatment plants (WWTPs) because the classical methods of wastewater treatment in WWTPs (mechanical, biological, chemical) do not completely remove pharmaceuticals and EDCs from the wastewater stream [11,12,13,14].

In recent years, some advanced technologies, such as advanced oxidation process (AOP), electrochemical oxidation, activated carbon adsorption, membrane techniques, and membrane bioreactors have been introduced for the removal of pharmaceuticals from wastewater [15,16,17,18]. However, these advanced processes are very expensive, and they are often unprofitable in technological systems. Therefore, the choice of cheaper, less complicated but very effective techniques for wastewater treatment with regard to pharmaceutical residues is well-aimed.

One of the options is to combine conventional WWTPs with hydroponic cultures called constructed wetlands (CWs). The basis of this is the use of plants such as macrophytes in the process of biological wastewater treatment [19]. In most cases, CWs are characterized based on water position and flow direction, and they are divided into surface flow (SF) systems, horizontal subsurface flow systems (H-SSF), and vertical subsurface flow systems (V-SSF). In the first case, sewage flows along a pool with growing plants. The root zone and part of the plants are located under the sewage stream. In H-SSF and V-SSF systems, sewage is introduced into a porous medium (generally gravel) in which macrophytes grow [20]. In most cases, constructed wetlands are a separate element of a sewage treatment plant. CWs act as a primary step (raw influent goes to the CW), secondary step (raw influent goes to primary treatment and afterwards to the CW), and tertiary step (raw influent goes to primary treatment, followed by secondary treatment and finally to the CW). This has been clearly and understandably described by Verlicchi et al. [20].

In recent years, constructed wetlands have been reported to be highly efficient in the removal of conventional pollutants from domestic sewage, agricultural sewage, industrial wastewater, mine drainage, leachate, contaminated ground water, and urban runoff [21,22,23,24,25,26]. Based on the literature, it is known that the mechanisms of pollution removal in CWs can be classified into biotic processes (e.g., microbiological degradation, root, and plant uptake) and abiotic processes like evaporation, photodegradation, oxidation, hydrolysis, retention, or sorption [27]. The efficiency of the removal of a pollutant depends also on such factors as seasonality, weather, and humidity. On the other hand, the use of constructed wetlands to remove pharmaceuticals and EDCs from wastewater is still not fully understood [28]. Most of the research concerning the uptake of pharmaceuticals and EDCs by plants in CWs refers to so-called microcosm, or pilot-scale laboratory systems, as opposed to real systems in WWTPs (Appendix A). In addition, only a few scientific papers have examined the actual uptake of emerging environmental pollutants by plants (Appendix A). Moreover, in most cases, the removal efficiency of the target compounds from wastewater is determined based only on determining their concentrations in raw and treated wastewater without the examination of their presence in plant tissues [29]. For this reason, it is difficult to assume the influence and mechanism of the removal of EDCs from wastewater by plants, as well as to establish which plants are the best for this purpose.

In this study, in order to obtain the most reliable data, we decided to evaluate the possibility of using hydroponic cultivation for the removal of pharmaceuticals and endocrine disrupting compounds in municipal sewage treatment, both with the application of real WWTP conditions as well as with the determination of the analytes studied in plant materials.

For this reason, the main objectives of this work were as follows: (1) to determine the selected pollutants in raw and treated sewage; (2) to develop an analytical method for determining selected pharmaceuticals and endocrine disrupting compounds in plant materials; (3) to assess the bioaccumulation of the mentioned pollutants in hydroponically cultivated plants; (4) to evaluate which of the tested plant species best meets this purpose.

According to literature data [30,31,32,33] and information included in Appendix A, for the isolation of pharmaceuticals and EDCs from plant materials, extraction techniques such as Microwave Assisted Extraction (MAE) [34], QuEChERS (Quick, Easy, Cheap, Effective, Rugged, and Safe) procedures [35,36], Accelerated Solvent Extraction (ASE) [27], Solid-Liquid Extraction (SLE) under gentle mixing [32], as well as Ultrasound-Assisted Extraction (UAE) [33] are used. In this study, we decided to check the usefulness of an ASE-Solid-Phase Extraction (SPE) technique for the isolation of target compounds from plant materials, because in none of the published papers, such investigations have been performed for the same target compounds and with the application of the same plants.

## 2. Results and Discussion

### 2.1. Determination of Selected Pharmaceuticals and Endocrine Disrupting Compounds in Treated and Untreated Sewage

The method for the determination of pharmaceuticals and EDCs in untreated and treated sewage was described in Section 3.4. The identification of each analyte was based on retention time, quantitative ion, and conformation ion/s, the quantification analysis on the area peak of quantitative ion (details are presented in Section 3.4). The mass spectra of target compounds with the MS fragments assignation are included in Appendix A. The determined concentrations of fifteen target compounds in untreated and treated sewage from a municipal wastewater treatment plant (MWWTP) are presented in Table 1.

Imipramine, diclofenac, and nadolol were not found in both types of samples (concentrations below the method detection limit (MDL) values). The concentrations of the analyzed NSAIDs (ibuprofen, naproxen) in untreated sewage were the highest among the investigated drugs (2695 ± 916 ng/L, 3420 ± 342 ng/L, respectively), which positively correlated with their large consumption and easy availability [6]. Amitriptyline, one of the measured antidepressant drugs, was found in a high concentration in untreated sewage (1676 ± 184 ng/L), and its concentration after wastewater treatment was still significant (613 ± 38 ng/L). Antidepressant drugs were identified in this area in 2014, and also in the Utrata River to which treated sewage is discharged [37].

In this study, for first time in this part of Europe, concentrations of estrogenic hormones in sewage derived from a wastewater treatment plant supported by constructed wetlands were investigated. E1, E2, E3, and synthetic EE2 were found in both types of sewage in concentrations ranging from 52 ± 3 ng/L to 1662 ± 260 ng/L in untreated sewage, and from 4 ± 0 ng/L to 122 ± 18 ng/L in treated sewage. The concentration of EE2 was the highest among the investigated estrogenic hormones, both in untreated and treated sewage (Table 1).

The concentrations of target compounds in untreated and treated sewage determined in this study were in the range presented by other authors for full-scan systems (Appendix A). For example, the concentrations of 86 pharmaceuticals, e.g., diclofenac, ibuprofen, ketoprofen, naproxen, EE2, E2, and E3 in untreated water were in the range from 1 ng/L to 1,000,000 ng/L, and in treated sewage to 901,618 ng/L [32].

However, for the first time, fifteen pharmaceuticals and EDCs were determined in untreated and treated sewage derived from a wastewater treatment plant supported by constructed wetlands working in a temperate climate zone.

### 2.2. Evaluation of the Analytical Method for Determining Target Compounds in Plant Materials

As previously mentioned, understanding the impact of individual plant species on the removal of pharmaceuticals and EDCs from wastewater requires an assessment of the uptake of these compounds by plants, for which an analytical procedure is required. In none of the published papers ([30,31,32,33], Appendix A), such investigations have been performed for the same target compounds and with the application of the same plants. For this reason, our investigations began from an evaluation of the analytical method for determining 15 target compounds in plant materials.

We decided to check the usefulness of the ASE technique for this purpose. ASE technique has been used only once for the isolation of 18 analytes [27]. In the work, among the tested analytes, were 7 pharmaceuticals (ibuprofen, ketoprofen, naproxen, diclofenac, salicylic acid, caffeine, carbamazepine), which were determined in *Typha angustifolia* and *Phragmites australis* plants (only roots) during an investigation of the behavior of pharmaceuticals and personal care products in mesocosm-scale constructed wetland compartments. A mixture of acetone/hexane (1:1; *v*/*v*), a temperature of 104 °C and two extraction cycles of 13.5 min were applied for ASE. A clean-up step was performed using a florisil column with different elution of the neutral/acidic fractions. In other studies based on other extraction procedures, for the isolation of pharmaceuticals from plant materials, different solvents such as methylene chloride:MeOH (2:1 *v*/*v*) and MTBE (Methyl *tert*-butyl ether):MeOH (1:1 *v*/*v*) [34], hexane:ethyl acetate (1:1, *v*/*v*) [35], 0.1 M HCl:ACN (1:1, *v*/*v*) [36], ACN with 0.5% formic acid (*v*/*v*) [32], and anhydrous sodium sulfate and MeOH [33] were applied.

In this study, the usefulness of three new extraction solvents: MeOH:H_2_O (1:1, *v*/*v*), EtOH:H_2_O (1:1, *v*/*v*), ACN:H_2_O (1:1, *v*/*v*), which were used at two temperatures: 50 °C and 80 °C for the isolation of target compounds by ASE was studied. Moreover, the effect of the acidification of the obtained extracts prior to SPE purification (pH ~ 2) was investigated (Figure 1).

The results of the optimization of the ASE procedure are presented in Table 2. As can be seen, among the tested ASE conditions, the best recoveries of target pharmaceuticals (ranging from 49 ± 6 to 111 ± 9%), were obtained during the application of a mixture of MeOH:H_2_O (1:1, *v*/*v*) and a temperature of 50 °C (Table 2). Only the recoveries of tri-cyclic antidepressant drugs (amitriptyline and clomipramine) were lower. It should be highlighted that these two compounds were not subjected to the derivatization procedure and the efficiency of ionization of these two compounds in an ion source during the GC-MS(SIM) analysis was high. In effect, the sensitivity of the proposed analytical method for the determination of these analytes was enough despite low recovery results.

The analyzed pharmaceuticals and EDCs contain polar functional groups (pKa are presented in Appendix A), therefore the acidification of ASE extracts prior to SPE purification could reduce or stop the dissociation of weak acids and increase the dissociation of weak bases. Oasis HLB cartridges used in the SPE procedure are stable from pH 0–14, and they are useful for the isolation of acidic, basic, and neutral analytes. However, modifications of the pH of the loaded sample could strongly influence the recovery data. For this reason, the effect of the acidification of ASE extracts to pH 2 prior to SPE purification on the recovery of analytes was also evaluated. Such studies were performed for the most effective extraction mixture (MeOH:H_2_O 1:1, *v*/*v*; Table 2) which was tested at two temperatures: 80 °C and 50 °C. The results are shown in Figure 2.

As we can see (Figure 2), in most cases, the acidification of ASE extracts led to a decrease in recoveries, both at 80 °C and 50 °C. It was observed for weak acids such as naproxen, difunisal, diclofenac), and weak bases (paracetamol, terbutaline, amitriptyline, clomipramine and nadolol; Appendix A). In case of acidification of the ASE extracts obtained at 80 °C (Figure 2A) the extraction efficiencies increased for ibuprofen, imipramine, estrone (E1), 17β-estradiol (E2), 17α-ethinylestradiol (EE2), and estriol (E3). However, when the ASE procedure was performed at 50 °C, these values decreased significantly for all mentioned compounds (Figure 2B). The extraction efficiencies of polar drugs: paracetamol (logP 0.46), terbutaline (logP 0.90), and nadolol (logP 0.81), decreased significantly where the ASE extracts obtained at 50 °C were subjected to acidification prior SPE (Figure 2B). Similar results were observed in our previous study concerning the development of a method for determining NSAIDs and natural estrogens in the mussel *Mytilus edulis trossulus* [38]. In summary, based on the obtained results, the application of the mixture of MeOH:H_2_O (1:1, *v*/*v*) at 50 °C without the acidification of the ASE extract prior to SPE purification was chosen as the most optimal ASE-SPE conditions for the isolation of target compounds from plant material.

In order to compare the recovery data obtained in this work with those presented by other scientists for pharmaceuticals and EDCs isolated from plants used in constructed wetlands, we tried to find the necessary information in all the cited papers [27,33,34,35,36]. Unfortunately, the recovery data were presented only in one of them [35], and only for triclosan, methyl-triclosan, and triclocarban. Nuel and coworkers [32] stated that drug detection and quantification methodologies had been fully described in their paper submitted to the journal Science of the Total Environment (Nuel et al. (2018)), but this paper was not published (based on data from Scopus; 31.10.2019).

### 2.3. Validation Parameters of the Proposed ASE-SPE-GC-MS(SIM) MetHod for Determining Target Compounds in Plant Materials

The ASE-SPE-GC-MS(SIM) methodology for the determination of target pharmaceuticals in plant samples has been validated in accordance with the guidelines described in Section 3.6. The determined validation parameters are presented in Table 3.

The coefficient of determination (R^2^) was in the range of 0.9760–0.9997; the intermediate precision measurement in the range of 0.05–21.60%. Mean recoveries were between 80% and 102% (Table 3). The lowest MDL value was recorded for ibuprofen, flurbiprofen, naproxen, diflunisal, diclofenac, and EE2 – 0.4 ng/g d.w.; the highest value of 2 ng/g d.w. was calculated for clomipramine and amitriptyline. Matrix effects were not significant for most of the analyzed compounds, apart from naproxen (ME 43%). The highest suppression of signals was observed for terbutaline (ME −31%) and flurbiprofen (ME −20%). Generally, matrix effect values were similar to those presented in our previous studies for the same target pharmaceuticals isolated from environmental matrices [39].

A comparison of the obtained matrix effect data with those presented by other authors was not possible because matrix effect values were not included in the cited papers [27,32,33,34,35,36]. The method quantification limit (MQL) and MDL values were similar to those we presented (Appendix A).

### 2.4. Assessment of Bioaccumulation of Selected Pollutants in Hydroponically Cultivated Plants

The determined concentrations of target compounds in three species of hydroponically cultivated plants (in ng/g dry weight) and the elimination efficiency of pharmaceuticals and EDCs in an MWWTP with constructed wetlands are presented in Table 4. The elimination efficiency (EE) was calculated according to Equation (6) described in Section 3.8.

Among the 15 investigated analytes, three NSAIDs (ibuprofen, flurbiprofen, diflunisal), two antidepressants (amitriptyline, imipramine), and one synthetic hormone (EE2) were found in Papyrus (*C. Papyrus*) (Table 4). In the case of the Yellow pimpernel (*L. nemorum*) species, terbutaline (β_2_-agonists), naproxen, and diflunisal as well amitriptyline and EE2 were found. The tissue of the European spindle (*E. europaeus*) plant contained ibuprofen, diflunisal, amitriptyline, and EE2. The highest concentrations were observed for diflunisal and terbutaline determined in Yellow pimpernel (*L. nemorum*) (5569 ± 1298 and 5323 ± 1869 ng/g d.w., respectively). The concentrations of other target compounds were below the MDL value (Table 4). Among three determined estrogenic hormones, only EE2 was found in all the investigated plant species; its concentration was the highest in Papyrus (*C. Papyrus*) (4126 ± 821 ng/g dry weight); E1 and E2 (natural hormones occurring in living organisms) were not detected in any plant tissues. Example chromatograms with marked SIM ions, recorded in this study for real plant samples, are presented in the Appendix A).

Few studies directly point to the transport or uptake of pharmaceuticals by plants in constructed wetlands (Appendix A). For example, 0.2% of the initial amount of diclofenac (1 mg/L) was detected in the roots and leaves of *Typha latifolia* during one week of exposure (laboratory system) [36]. Hijosa-Valsero et al. [27] investigated the possibility of the uptake of 18 selected analytes, including ibuprofen, ketoprofen, naproxen, diclofenac, salicylic acid, caffeine, and carbamazepine by *Typha angustifolia* and *Phragmites australis* plants in constructed wetlands (mesocosm-scale). Ibuprofen, salicylic acid and caffeine were detected in plant tissues, whereas ketoprofen, naproxen, diclofenac and carbamazepine were not found in these species. The main substance detected in root tissues was salicylic acid (123–2560 ng/g). The authors confirmed that *T. angustifolia* is more suitable for the removal of the investigated pollutants than *Phragmites australis* [27]. In another study, performed by Nuel et al. [32], ibuprofen was found in all the investigated plant samples (*Salix alba*, *Iris pseudacorus*, *Juncus effusus*, *Callitriche palustris*, *Carex caryophyllea*) used in constructed wetlands. According to Wang et al. [33] the bioconcentration factors (BCFs) in *Typha angustifolia* of 8 compounds, e.g., caffeine, carbamazepine, ibuprofen, fluoxetine, gemfibrozil ranged between 60 and 2000. Concentrations of the determined compounds in plant tissue were up to several hundred ng/g for caffeine. This confirms that the results obtained in this study are in agreement with the literature data. However, for the first time, such investigations were performed for Papyrus (*Cyperus papyrus*), European spindle (*Euonymus europaeus*) and Yellow pimpernel (*Lysimachia nemorum*) taking into account this selected group of pharmaceuticals and EDCs.

In the study, the obtained elimination efficiency (EE) of the investigated compounds from wastewater was in the range of 35.8% for diflunisal to above 99.9% for paracetamol, terbutaline and flurbiprofen (Table 4). For comparison, the literature EE data for ibuprofen, presented in Appendix A**,** are as follows: 5–88% [40], 96% [41,42,43], 44–77% [44], 94% [45], 52–85% [46], >99% [47], 51–54% (winter), 85–96% (summer) [48], 42–99% [49]. In our study, this was 99.7%, which confirms that the elimination efficiency of ibuprofen was comparable or higher than that calculated for other studies. Similar the EE data for pharmaceuticals is shown also in papers [50,51,52]. Thus, the obtained results proved the uptake of the investigated micropollutants by plants allowed an increase in the effectiveness of their removal from wastewater in such a system.

### 2.5. Assessment of the Usefulness of Hydroponically Cultivated Plants for Removing Target Compounds from Sewage Stream

According to the literature data, due to the considerable size of rhizomes and roots, the most frequently hydroponically cultivated plants are *Typha sp.* and *Phragmites sp.* (Appendix A). In this study, the possibility of using three plants: *Cyperus papyrus* (Papyrus), *Lysimachia nemorum* (Yellow pimpernel), and *Euonymus europaeus* (European spindle) for this purpose was examined for the first time. These plant species are very well adapted to growth in MWWTPs and exhibit the strongest growth during the growing season. In order to assess the usefulness of these species for removing pharmaceuticals and EDCs from the sewage stream, the sum of the masses of all the target compounds taken by the tested species was established and the obtained values are presented in Table 5.

Therefore, the highest concentration of all target compounds was observed for *the Lysimachia nemorum* (Yellow pimpernel) plant (16,849 ng/g d.w.), followed by *Cyperus papyrus* (Papyrus) (12,439 ng/g d.w.), and the lowest for *Euonymus europaeus* (European spindle) (2525 ng/g d.w.). Taking into account the summary uptake of target pharmaceuticals and EDCs by tested plants, the *Lysimachia nemorum* (Yellow pimpernel) species is the best for this purpose. The differences in the uptake of target compounds by these plants are connected to their morphological structures. *Euonymus europaeus* (European spindle) has a significant number of woody stems and rhizomes, which means that the uptake of target compounds by this plant is much lower than by the green parts of the two other tested species. Thus, the determination of target compounds in untreated wastewater and treated wastewater, as well as in plant tissues, allowed for establishing which hydroponic cultivation system supported the wastewater treatment process the most significantly (Table 3, Table 4 and Table 5).

## 3. Materials and Methods

### 3.1. Chemicals and Materials

Pure standards (>98%) of ibuprofen, paracetamol, flurbiprofen, naproxen, diflunisal, diclofenac sodium salt, nadolol, terbutaline, amitriptyline, imipramine, clomipramine, estrone (E1), 17β-estradiol (E2), 17α-ethinylestradiol (EE2), and estriol (E3), as well as the derivatization reagent *N,O*-bis(trimethylsilyl)trifluoroacetamide (BSTFA) containing 1% of trimethylchlorosilane (TMCS) and pyridine (99.8%) were purchased from Sigma-Aldrich (Steinheim, Germany). Solvents: methanol (MEOH), ethanol (EtOH), acetonitrile (ACN) were supplied by POCH (Gliwice, Poland), ethyl acetate (EtOAc) by Sigma-Aldrich, while 37% hydrochloric acid (HCl) for the acidification of extracts was provided by Chempur (Piekary Śląskie, Poland). Solid-phase extraction (SPE) was performed using Oasis HLB cartridges (6 mL, 200 mg, Waters Corporation, Miliford, USA).

Standard stock solutions of target compounds (1 mg × mL^−1^) were prepared in methanol. All the stock solutions were stored at –20 °C. Working calibration standard solutions were prepared by diluting the standard stock solutions in the appropriate amount of methanol, and they were stored in the dark at –20 °C.

### 3.2. MWWTP and Constructed Wetlands Characteristic

The research was carried out at the municipal wastewater treatment plant (MWWTP) in Sochaczew (Mazowieckie Voivodeship, Poland). In Sochaczew MWWTP, macrophyte cultures are introduced in combination with an oxygen biological reactor. The efficiency of removing conventional pollution in Sochaczew MWWTP is high, as presented in Table 6.

Sochaczew MWWTP consists of the following elements: 1° mechanical wastewater treatment (grates with a throughput of 515 m^3^/h, aerated sand traps with degreasers, aerated at 1.91 m^3^/min), 2° biological wastewater treatment (flow reactor with activated sludge with throughput at 6000 m^3^/d and secondary settling tank with active capacity at 1142 m^3^), sediment trap settling tanks with a diameter of 21 m, excessive sludge dewatering station, and lime treatment. The wastewater treatment plant was designed for 55,925 equivalent number of inhabitants with a maximum daily volume of sewage at 11,636 m^3^/d.

Constructed wetlands have been implemented in the second stage of wastewater treatment (biological) as a wastewater polishing treatment system. Wastewater mixed with activated sludge flows around plants. Contact between the plants and wastewater occurs only in the rhyzophytic zone (Appendix A). In this way, the green parts do not have contact with the walls and the benefits are the lack of dieback of the shoots, and faster acclimatization. The plants used in the constructed wetlands are as follows: Papyrus (*Cyperus papyrus*), Reed (*Phragmites australis* (Cav.) Trin. ex Steud, 1841), Spathiphyllum (*Spathiphyllum*
*Adans.*), Grey willow (*Salix cinerea*), Rushes (*Juncus tenageia* Ehrh.), Sweet flag (*Acorus calamus*), Yellow iris (*Iris pseudacorus*), European spindle (*Euonymus europaeus*), Yellow pimpernel (*Lysimachia nemorum*), and Summer lilac (*Buddleja davidii* Franch).

### 3.3. Sampling Sewage and Plants from Constructed Wetland

The samples of untreated and treated sewage were collected in November 2017 into 2.5 L amber glass bottles. Raw sewage samples were taken before mechanical treatment, and treated sewage samples were collected at the outlet to the Utrata River. In the laboratory, the sewage samples were filtered under pressure using a 0.45 μm nylon filter and frozen at −20 °C until analysis. A list of the monitored pharmaceuticals and EDCs with physical and chemical properties is included in the Appendix A.

At the same time, samples of the plants used in the constructed wetlands were collected. Only parts which had no contact with sewage were collected to confirm the migration of pharmaceuticals to green tissue. The plants were washed under tap water, separated and dried for 3 days at room temperature (~23 °C). Finally, the plants were dried at 60 °C for 3 hrs in a heating oven (Pol-Eko Apparatures sp.j., Wodzisław Śląski, Poland). The dried plants were homogenized using a mechanical blender (Kenwood, Havant, UK), and frozen at −20 °C until analysis. The average water content in *Cyperus papyrus*, *Lysimachia nemorum* and *Euonymus europaeus*, determined based on the weight of the sample before and after desiccation, was 75.4%, 64.7%, and 68.5%, respectively.

### 3.4. Determination of Selected Pharmaceuticals and Endocrine Disrupting Compounds in Treated and Untreated Sewage

Pharmaceuticals and endocrine disrupting compounds were extracted from treated and untreated sewage using SPE on Oasis HLB cartridges (3 mg, 6 mL) according to a previously optimized and validated procedure [53]. Each cartridge was preconditioned with 3 mL of EtOAc, 3 mL of MeOH, and 3 mL of distilled water adjusted to pH 2 with 1 M HCl. A liquid sample (250 mL of wastewater adjusted to pH 2 with 1 M HCl) was passed through a cartridge at a flow rate of ~5 mL min^−1^ using a vacuum manifold. After loading the sample, the cartridge was flushed with 10 mL of an MeOH:H_2_O (1:9, *v*/*v*) mixture and subsequently air-dried under a vacuum for 30 min. The adsorbed analytes were eluted with 6 mL of MeOH and dried completely under nitrogen gas in order to be derivatized. Derivatization was carried out using 100 µL of a mixture of 99% BSTFA/1% TMCS and pyridine (1:1, *v*/*v*) [53]. After shaking for 30 s, the reaction vials were placed in a heating block for 30 min at 60 °C. The standards of pharmaceuticals were derivatized using the same procedure. The samples were analyzed using the GCMS-QP 2010 SE Shimadzu System (Shimadzu, Kyoto, Japan) with an AOC-5000 autosampler. The separation of analytes was done using a Zebron ZB-5MSi fused-silica capillary column (30 m, 0.25 mm I.D., 0.25 µm film thickness, Phenomenex). The injection port temperature was 300 °C and 1 µL samples were injected in the splitless mode (1 min). The carrier gas was helium (100 kPa). The oven temperature programme was: 120 °C for 1 min, from 120 °C to 300 °C at 6 °C/min, and finally, 4 min at 300 °C (total 35 min). The transfer line was held at 300 °C. The scan time was 0.3 s. The parameters used for identifying analytes: retention time, quantitative ions, and conformation ions, and applied time windows for selected analytes are shown in Table 7.

### 3.5. Development of the Analytical Method for Determining Target Compounds in Plant Materials

An Accelerated Solvent Extraction (ASE) technique with purification of the obtained extracts by SPE was applied for extracting the pharmaceuticals and EDCs from plant tissues. The ASE extraction was performed using a DIONEX ASE 350 (Dionex Corp., Sunnyvale, CA, USA). Three solvent mixtures (MeOH:H_2_O 1:1 *v*/*v*; EtOH:H_2_O 1:1 *v*/*v*: ACN:H_2_O 1:1 *v*/*v*) were applied at two different temperatures (50 °C and 80 °C) for the ASE extraction of the analytes. Moreover, the obtained extracts, prior to SPE purification, were acidified to pH ~2 as well as being investigated without acidification. In each tested condition, the concentration of the target pharmaceuticals and EDCs in spiked plant samples was 1000 ng/g d.w.

A cellulose filter (19.8 mm, Dionex Corp.) was placed on the bottom of a 33-mL stainless steel extraction cell. The cell was filled with 3 ± 0.01 g of diatomaceous earth (on the cellulose filter). Then, 1 ± 0.01 g of spiked and non-spiked homogenous plant tissues was added. Next, the dead volume of the cell was filled with diatomaceous earth. Each extraction was carried out at constant parameters: heating time 5 min, static time 3 min in 3 cycles, and a 10% purge of 50 s. The extraction pressure was set to 1500 psi. The total extraction time was 18.5 min and the extraction volume was 26 ± 0.1 mL. A total of 5 mL of ASE extract was dissolved in deionized water and subjected to purification using SPE under the conditions presented in Section 3.4. A minimum of three samples for each tested parameter were prepared, and each sample was analyzed a minimum of three times.

### 3.6. Validation of the Proposed Method for Determining Target Compounds in Plant Species

The validation of the analytical method for determining target compounds in plant species was carried out using the matrix-matched calibration solutions and working calibration standard solutions in accordance with the guidelines of the International Vocabulary of Metrology [54]. The matrix-matched calibration solutions were prepared by spiking samples with eight different concentrations of pharmaceuticals and endocrine disrupting compounds in a range of 19.5–2500 ng/g d.w. Calibration curves were based on the external method, and they were obtained by plotting the peak area for each analyte against its concentration. The linearity, correlation coefficient (R^2^), intermediate precision measurement (expressed by RSD, *n* = 3), and mean recovery (MR) were established according to procedures described in our previous paper [55]. Briefly, the mean recovery of the analytical method was determined based on the known concentrations of target compounds in the tested samples (C_known_) and the concentrations determined by the analysis (C_determined_) using Equation (1):**MR** = (C_determined_/C_known_) × 100%(1)

The method quantification limit (MQL) was determined using Equation (2):**MQL** = (IQL × 100%)/(CF × AR)(2)
whereas the method detection limit (MDL) using Equation (3):**MDL** = MQL/3(3)
where IQL is the instrumental quantification limit, CF is the concentration factor, and AR is the absolute recovery of analytes (%).

The matrix effect (ME) and absolute recovery (AR) were determined according to procedures described by Caban et al. [56] using Equation (4) and Equation (5), respectively:**ME** = ((B − D/A) − 1) × 100%(4)
**AR** = ((C − D)/A) × 100%(5)
where A is the peak area of the analyte recorded for the standard solution, B is the peak area of the analyte recorded for the sample spiked with the target compound after extraction, C is the peak area of the analyte recorded for the sample spiked with the target compound before extraction, and D is the peak area of the analyte recorded for the non-spiked sample (blank sample).

### 3.7. Application of the Proposed Method for Determining Target Compounds in Hydroponically Cultivated Plants

The optimized and validated ASE-SPE-GC-MS(SIM) method was used to determine 15 target compounds in 3 species of plants (Papyrus (*Cyperus papyrus*), European spindle (*Euonymus europaeus*), Yellow pimpernel (*Lysimachia nemorum*)) used in the constructed wetlands in Sochaczew MWWTP. These species were chosen due to their best adaptation to growth in the MWWTP. In addition, the mentioned plants exhibit the strongest growth during the growing season, which affects the transport of pollutants to plant tissues.

### 3.8. Evaluation of the Effectiveness of Removing Pharmaceuticals and Endocrine Disrupting Compounds in MWWTP Sochaczew

Concentrations of pharmaceuticals and EDCs in treated and untreated sewage were used to determine the elimination efficiency (EE%) from the wastewater stream in the MWWTP. The elimination efficiency factor was calculated using Equation (6):**EE%** = (C_untreated_ − C_treated_)/(C_untreated_) × 100%(6)
where C_untreated_ is the measured concentration of the pharmaceutical in untreated sewage samples, and C_treated_ is the measured concentration of the pharmaceutical in treated sewage samples. This parameter allowed for establishing the effectiveness of removing pharmaceuticals in a municipal treatment plant supported by constructed wetlands.

## 4. Conclusions

In this study, the analysis of the possibility of using hydroponic cultivation for the removal of 15 pharmaceuticals and endocrine disrupting compounds in municipal wastewater treatment plants is presented. For the first time, three plants: *Cyperus papyrus* (Papyrus), *Lysimachia nemorum* (Yellow pimpernel), and *Euonymus europaeus* (European spindle) were considered. In order to obtain the most reliable data, the investigation was performed using real MWWTP conditions and with the determination of target compounds not only in raw and treated wastewater, but also in plant materials. The determination of target compounds in raw and treated wastewater samples was performed using a previously proposed method [53]; however, for determining target compounds in plant materials, in this study, a new ASE-SPE-GC-MS(SIM) method was developed and validated. The application of a mixture of MeOH:H_2_O (1:1, *v*/*v*) at 50 °C without the acidification of the ASE extract prior to SPE purification was found to be the most optimal ASE-SPE conditions for the isolation of target compounds from plant material. The MQL values of the proposed method were in the range of 0.4 ng/g d.w. to 2 ng/g d.w.; the intermediate precision measurement was in the range of 0.05 to 21.60%; the mean recoveries in the range of 80% to 102%. Among the 15 investigated, 5 analytes were found in Papyrus (*C. Papyrus*); 5 target compounds in Yellow pimpernel (*L. nemorum*), and 4 in the tissue of the European spindle (*E. europaeus*) plant. The highest concentration of all target compounds was observed for *Lysimachia nemorum* (Yellow pimpernel), therefore, taking into account the summary uptake of target pharmaceuticals and ECDs by the tested plants, this species is the best for supporting conventional MWWTPs. The obtained data proved that the elimination efficiency of the investigated compounds from wastewater was in the range of 35.8% to 100%. Thus, the application of constructed wetlands for supporting conventional MWWTPs allowed a significant increase in their removal from the wastewater stream.

Establishing which plants effectively cumulate certain types of compounds is useful for designing effective constructed wetlands. Moreover, in the future, the proposed method for determining pharmaceuticals and EDCs in plant materials could be used for assessing the quality of food of plant origin in the cultivation of which sewage sludge or purified sewage was used.

## Figures and Tables

**Figure 1 molecules-25-00162-f001:**
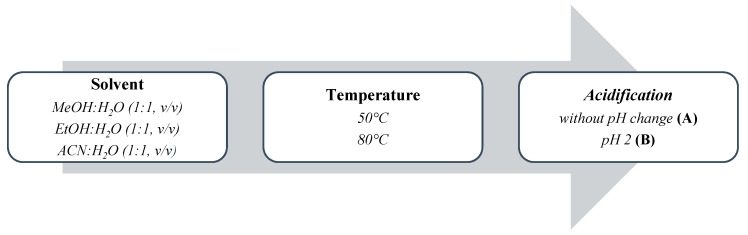
Selection of the optimal Accelerated Solvent Extraction (ASE)-SPE conditions for the isolation of target compounds from plant materials.

**Figure 2 molecules-25-00162-f002:**
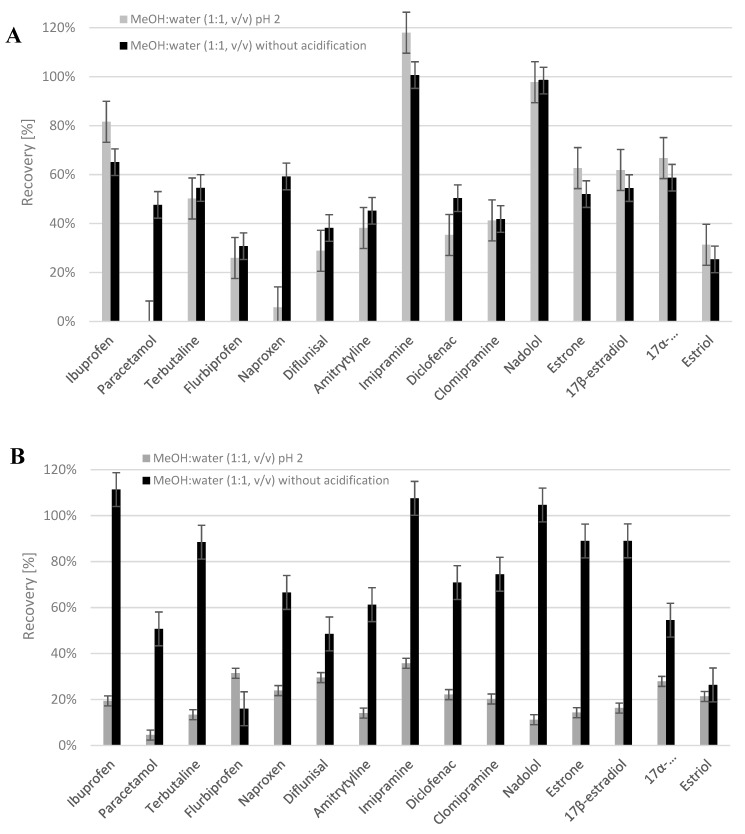
Impact of the acidification of ASE extracts prior to SPE purification on the recoveries of analytes, carried out at 80 °C (**A**) and 50 °C (**B**), respectively.

**Table 1 molecules-25-00162-t001:** Concentrations of target compounds in untreated and treated sewage samples collected from a municipal wastewater treatment plant (MWWTP), determined using the Solid-Phase Extraction (SPE)-GC-MS(SIM) method (*n* = 3).

Pharmaceuticals	Concentration in Untreated Sewage	Concentration in Treated Sewage
	(mean ± SD) [ng/L]	
**Ibuprofen**	2695 ± 916	8 ± 1
**Paracetamol**	2130 ± 298	<MDL
**Terbutaline**	154 ± 8	<MDL
**Flurbiprofen**	51 ± 12	<MDL
**Naproxen**	3420 ± 342	38 ± 1
**Diflunisal**	67 ± 12	43 ± 7
**Amitriptyline**	1676 ± 184	613 ± 38
Imipramine	<MDL	<MDL
Diclofenac	<MDL	<MDL
**Clomipramine**	169 ± 29	50 ± 5
Nadolol	<MDL	<MDL
**Estrone**	52 ± 3	19 ± 2
**17β-estradiol**	110 ± 13	4 ± 0
**17α-ethinylestradiol**	1622 ± 260	122 ± 18
**Estriol**	273 ± 27	13 ± 3

**Table 2 molecules-25-00162-t002:** Recoveries (mean ± SD) of fifteen analytes using different conditions of the ASE procedure (*n* = 3, conc. 1000 ng/g d.w.).

Type of Solvents/ Temperature/ Pharmaceuticals	EtOH:H_2_O (1:1, *v*/*v*)		ACN:H_2_O (1:1, *v*/*v*)		MeOH:H_2_O (1:1, *v*/*v*)	
	80 °C	50 °C	80 °C	50 °C	80 °C	50 °C
	Value of Recovery [% ± SD (%)]					
**Ibuprofen**	105 ± 22	154 ± 5	105 ± 16	194 ± 13	65 ± 5	111 ± 9
**Paracetamol**	26 ± 11	36 ± 7	10 ± 2	25 ± 3	42 ± 5	74 ± 9
**Terbutaline**	16 ± 5	25 ± 6	19 ± 4	26 ± 3	48 ± 3	51 ± 5
**Flurbiprofen**	89 ± 15	120 ± 18	80 ± 14	140 ± 8	52 ± 2	89 ± 10
**Naproxen**	91 ± 9	118 ± 6	80 ± 13	139 ± 7	54 ± 3	89 ± 11
**Diflunisal**	90 ± 18	115 ± 7	90 ± 11	159 ± 7	55 ± 3	88 ± 12
**Amitriptyline**	27 ± 8	58 ± 11	52 ± 11	10 ± 3	25 ± 4	26 ± 5
**Imipramine**	29 ± 8	88 ± 25	74 ± 10	16 ± 2	59 ± 2	55 ± 8
**Diclofenac**	103 ± 12	10 5± 3	68 ± 8	76 ± 9	98 ± 6	105 ± 5
**Clomipramine**	16 ± 7	44 ± 19	60 ± 8	6 ± 1	31 ± 6	16 ± 4
**Nadolol**	73 ± 6	91 ± 12	59 ± 4	97 ± 17	59 ± 3	67 ± 9
**Estrone**	63 ± 13	80 ± 15	49 ± 6	67 ± 5	38 ± 3	49 ± 6
**17β-estradiol**	77 ± 10	86 ± 15	71 ± 11	100 ± 5	45 ± 4	61 ± 8
**17α-ethinylestradiol**	125 ± 16	130 ± 10	235 ± 14	219 ± 12	101 ± 9	108 ± 10
**Estriol**	65 ± 7	93 ± 14	64 ± 7	88 ± 7	50 ± 4	71 ± 7

**Table 3 molecules-25-00162-t003:** Selected validation parameters of the ASE-SPE-GC-MS(SIM) method for determining target compounds in plant samples (analytical range from method quantification limit (MQL) to 2500 ng/g). Precision and accuracy were determined for three concentrations from analytical range (78 ng/g d.w., 625 ng/g d.w., and 2500 ng/g d.w.).

Validation Parameters	Calibration Curves	R^2^	Measurement Intermediate Precision (RSD%)	Mean Recovery	ME	MQL	MDL
Compound				(%)	%	(ng/g d.w.)	
**Ibuprofen**	68698 (±635) x + 1208.4 (±648.4)	0.9995	0.44–7.61	80–102	10	0.4	0.1
**Paracetamol**	74010 (±4738) x − 8672.6 (±4835.8)	0.9760	2.46–20.25	100	19	0.5	0.2
**Terbutaline**	364393 (±2459) x − 1622.6 (±2510.1)	0.9997	0.39–8.75	99–100	−31	0.8	0.3
**Flurbiprofen**	67739 (±632) x − 737.2 (621.9)	0.9996	0.46–10.86	95–98	−20	0.4	0.1
**Naproxen**	67687 (±829) x − 1444.7 (±880.5)	0.9994	3.73–13.65	99–101	43	0.4	0.1
**Diflunisal**	114510 (±2633) x − 4994.1 (±2590.4)	0.9970	0.05–9.33	97–102	15	0.4	0.1
**Amitriptyline**	339801 (±3704) x + 30042.5 (3642.7)	0.9994	0.44–16.20	94–102	−9	2	0.7
**Imipramine**	33490 (±333) x − 752.78 (±339.8)	0.9994	1.08–14.00	97–100	12	0.7	0.2
**Diclofenac**	18224 (±287) x − 673.8 (±292.7)	0.9990	3.62–21.60	89–101	25	0.4	0.1
**Clomipramine**	17286 (±128) x − 26.9 (±125.7)	0.9997	0.56–17.09	99–100	−8	2	0.7
**Nadolol**	550924(±13936) x − 31,586 (±13706.5)	0.9970	1.82–9.86	91–102	28	0.6	0.2
**Estrone (E1)**	24502(±587) x + 284.4 (±599.1)	0.9970	2.03–14.88	99–100	−2	0.8	0.3
**17β-estradiol (E2)**	44380 (±1275) x − 276.4 (1302.1)	0.9950	0.18–10.78	80–101	−3	0.6	0.2
**17α-ethinylestradiol (EE2)**	12263 (±246) x + 560.9 (±251.5)	0.9980	0.16–12.16	80–102	−11	0.4	0.1
**Estriol (E3)**	9240 (±265) x + 724.7 (±303.3)	0.9970	1.00–15.78	99–100	31	0.5	0.2

**Table 4 molecules-25-00162-t004:** Results of determining target compounds in three species of hydroponically cultivated plants from an MWWTP using the developed ASE-SPE-GC-MS(SIM) method (*n* = 3), and the elimination efficiency of these compounds from wastewater in an MWWTP supported by constructed wetlands.

Pharmaceuticals	*Cyperus papyru* (Papyrus)	*Lysimachia nemorum* (Yellow Pimpernel)	*Euonymus europaeus* (European Spindle)	EE
	(mean ± SD) [ng/g Dry Weight]			%
**Ibuprofen**	700 ± 28	<MDL	1616 ± 124	99.7
**Paracetamol**	<MDL	<MDL	<MDL	>99.9
**Terbutaline**	<MDL	5323 ± 1869	<MDL	>99.9
**Flurbiprofen**	2107 ± 92	<MDL	<MDL	>99.9
**Naproxen**	<MDL	1422 ± 216	<MDL	98.9
**Diflunisal**	1569 ± 321	5569 ± 1298	260 ± 4	35.8
**Amitriptyline**	404 ± 5	1399 ± 20	435 ± 5	63.4
**Imipramine**	3533 ± 198	<MDL	<MDL	-^1^
**Diclofenac**	<MDL	<MDL	<MDL	-^1^
**Clomipramine**	<MDL	<MDL	<MDL	70.4
**Nadolol**	<MDL	<MDL	<MDL	-^1^
**Estrone (E1)**	<MDL	<MDL	<MDL	63.5
**17β-estradiol (E2)**	<MDL	<MDL	<MDL	96.4
**17α-ethinylestradiol (EE2)**	4126 ± 821	3136 ± 599	214 ± 3	92.5
**Estriol (E3)**	<MDL	<MDL	<MDL	95.2

^1^ if drug concentrations were below the MDL value in both untreated and treated wastewater, the elimination efficiency (EE) was not calculated.

**Table 5 molecules-25-00162-t005:** The sum of the uptake of selected pharmaceuticals and endocrine disrupting compounds (EDCs) in ng/g dry weight by tested plant species grown in an MWWTP.

Plant Species	*Cyperus papyru*	*Lysimachia nemorum*	*Euonymus europaeus*
**∑_Selected pharmaceuticals and ECDs_**	[ng/g dry weight]		
	12,439	16,849	2525

**Table 6 molecules-25-00162-t006:** List of sewage test results in the Municipal Wastewater Treatment Plant in Sochaczew (average values from 2017).

Factor	Results for Treated Wastewater	Results for Untreated Wastewater	The Highest Acceptable Concentration	Removal Efficiency
**BOD_5_**	3.4 mg/L O_2_	460 mg/L O_2_	15 mg/L O_2_	99.3%
**COD_Cr_**	32.7 mg/L O_2_	1144.7 mg/L O_2_	125 mg/L O_2_	97.1%
**N_total_**	8.8 mg/L	96.7 mg/L	15 mg/L	90.5%
**P_total_**	0.2 mg/L	11.9 mg/L	2 mg/L	98.8%
**Suspensions**	8.6 mg/L	567.5 mg/L	35 mg/L	98.5%
**pH**	7.2	7.9	-	-

BOD_5_ biochemical oxygen demand (for 5 days). COD_Cr_ Chemical Oxygen Demand (using Potassium Dichromate).

**Table 7 molecules-25-00162-t007:** Retention parameters (time allowed change ± 0.15 min), time windows, and SIM ions for trimethylsilyl (TMS)-derivatives of target compounds (quantitative ions are marked in bold; confirmation ions with intensity relative to quantitative ions (%)).

Identification Parameters/ Pharmaceuticals	Retention Time (R_t_) [min]	Characteristic Ions (*m/z*) (Quantitative and Confirmation Ions)	Time Windows [min]
**Ibuprofen**	10.39	**160**; 263(18); 278(10)	10.01–13.52
**Paracetamol**	10.64	**206**; 280(90); 295(60)	
**Terbutaline**	16.20	**86**; 356(47)	13.52–19.88
**Flurbiprofen**	17.01	**180**; 165(85); 301(25)	
**Naproxen**	18.36	**185**; 243(80)	
**Diflunisal**	18.88	**379**; 380(30)	
**Amitriptyline**	20.64	**58**; 215(30)	19.88–22.50
**Imipramine**	21.09	**243**; 58(60); 195(40)	
**Diclofenac**	21.73	**214**; 242(50); 352(12); 367(25)	
**Clomipramine**	23.90	**268;** 314(25)	22.50–25.63
**Nadolol**	24.53	**86**; 73(8); 510(33)	
**Estrone (E1)**	26.48	**342**; 257(50); 285(36); 327(10)	25.63–27.97
**17β-estradiol (E2)**	27.11	**416**; 285(90)	
**17α-ethinylestradiol (EE2)**	28.59	**425**; 440(35)	27.97–29.30
**Estriol (E3)**	29.77	**504**; 311(81)	29.30–30.39

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
