# Peer review of "Evaluation of the Possibility of Using Hydroponic Cultivations for the Removal of Pharmaceuticals and Endocrine Disrupting Compounds in Municipal Sewage Treatment Plants"

_molecules, 2019, doi:10.3390/molecules25010162_

Round 1
Reviewer 1 Report
I reviewed the manuscript “Evaluation of the possibility of using hydroponic cultivations for the removal of pharmaceuticals and endocrine disrupting compounds in municipal sewage treatment plants” (molecules-654125). The idea is interesting from analytical chemistry view, but the manuscript is focused on the environmental part (introduction and results).
The authors must emphasize the design and validation of the methodology employed to analyze pharmaceutical compounds in Papyrus, Yellow pimpernel and European spindle. On the other hand, the manuscript is suitable for an environmental journal.
Some points:
Tittle. The tittle must be changed in order to adequate it. Something like, “Determination of pharmaceutical and endocrine disrupting compound in hydroponic cultivations”
Abstract. It is not adequate to report an efficiency of 100%, it must be considered the limit of detection to establish the maximum efficiency achieved.
Please modify the introduction section in order to be suitable for the journal Scope.
Results and Discussion
Table 1, please include the concentration used for recovery studies.
Please include a deeper discussion about the effect of polarity, and acidification. It must be performed the Table A2.
Table 2, please include the confidence intervals for the analytical parameters in the calibration Curve. Include the concentration used for precision and accuracy determination. The values of analytical sensitivity are too high, please mention the internal standard used during the analysis.
Table 7. It could be interesting to include the MS fragments assignation, not just mention the ions selected.
Author Response
Dear Reviewer,
We would like to thank for valuable comments and suggestions. They will definitely help improve the quality of the manuscript. All of them have been taken into consideration. Please find enclosed point by point answers to these comments:
Reviewer’s 1 comments
General comment: I reviewed the manuscript “Evaluation of the possibility of using hydroponic cultivations for the removal of pharmaceuticals and endocrine disrupting compounds in municipal sewage treatment plants” (molecules-654125). The idea is interesting from analytical chemistry view, but the manuscript is focused on the environmental part (introduction and results).
The authors must emphasize the design and validation of the methodology employed to analyze pharmaceutical compounds in Papyrus, Yellow pimpernel and European spindle. On the other hand, the manuscript is suitable for an environmental journal.
Answer: In the revised manuscript, we significantly improved the presentation of the design and validation of the methodology employed to analyze pharmaceutical compounds in Papyrus, Yellow pimpernel and European spindle. We are fully agree with the Reviewer that the manuscript is also suitable for an environmental journal. It was submitted to the Special Issue of Molecules called “"Pharmaceutical Residues in the Environment" and fits very well to the thematic of this Special Issue. We have send the title and the short abstract of this manuscript to Editor of Molecules much earlier before submission to the journal (in April 2019), and the topic has been accepted by Editorial Board.
Specific comments:
The tittle must be changed in order to adequate it. Something like, “Determination of pharmaceutical and endocrine disrupting compound in hydroponic cultivations”Answer: As was mentioned, the title of the manuscript “Evaluation of the possibility of using hydroponic cultivations for the removal of pharmaceuticals and endocrine disrupting compounds in municipal sewage treatment plants” has been accepted by Editorial Board much earlier before submission of the manuscript to the journal (in April 2019). In our opinion, it is adequate to the study described in the manuscript.. However, if the modification of the title into version proposed by Reviewer, will be crucial for publication we will modify into the following one: “Determination of pharmaceutical and endocrine disrupting compounds in hydroponic cultivations being a part of municipal sewage treatment plants”
It is not adequate to report an efficiency of 100%, it must be considered the limit of detection to establish the maximum efficiency achieved.Answer: In the revised manuscript, the limits of detection of Paracetamol, Terbutaline, and Flurbiprofen were considered to establish the maximum efficiencies, and now these values are presented as above 99.9%. These values were corrected also in Table 6.
Please modify the introduction section in order to be suitable for the journal Scope.Answer: According to this suggestion, the introduction section was modified in order to be more suitable for the Molecules scope.
Table 1, please include the concentration used for recovery studies.Answer: According to this suggestion, the concentration used for recovery studies was introduced in the caption of Table 1.
Please include a deeper discussion about the effect of polarity, and acidification. It must be performed the Table A2.Answer: According to the Reviewer’s suggestion more deep discussion about the effect of polarity, and acidification is presented in Section 2.1.
Table 2, please include the confidence intervals for the analytical parameters in the calibration Curve. Include the concentration used for precision and accuracy determination. The values of analytical sensitivity are too high, please mention the internal standard used during the analysis.Answer: In the revised manuscript, the confidence intervals for the analytical parameters in the calibration curve are presented in Table 2. Precision and accuracy were determined for three concentrations from analytical range (78 ng/g d.w., 625 ng/g d.w. and 2500 ng/g d.w.). Now this information is included in the legend of Table 2. We would like to explain that the calibration curves were based on the external method, and they were obtained by plotting the peak area for each analyte against its concentration. Thus, the internal standard was not used during the analysis. For this reason, the values of analytical sensitivity (a parameters in calibration curves) are high.
Table 7. It could be interesting to include the MS fragments assignation, not just mention the ions selected.Answer: The MS fragments assignation is now included in Appendix A in Figure 3A and information about it is included in Section 3.4.
We hope that our explanations will be satisfactory for you and the revised manuscript could be accepted for publication in Molecules.
Yours sincerely,
Jolanta Kumirska

Reviewer 2 Report
The paper byKumirska et al. presents a thorough study on a procedure of wide environmental interest. The work is well planned and well conducted, the analytical procedure is accurately checked, and the results are surely significant. The paper is well presented and the conclusion are sound. In my opinion it can be accepted in the present form.
Author Response
Dear Reviewer,
We would like to thank for the comment. Please find enclosed answer to this comment.
Reviewer 2:
The paper by Kumirska et al. presents a thorough study on a procedure of wide environmental interest. The work is well planned and well conducted, the analytical procedure is accurately checked, and the results are surely significant. The paper is well presented and the conclusion are sound. In my opinion it can be accepted in the present form.
Answer: We would like to thank the Reviewer for this kind opinion, and we believe that our research will be interesting for the readers of Molecules journal.
Yours sincerely,
Jolanta Kumirska

Reviewer 3 Report
Lines 68-70 and 89-90 repeats the same information.
Lines 139-140: do not put in different lines value and units (50 ºC)
Figure 2: The y scale should start at zero, the inner legend of the graphs should be improved.
Table 2: Verify the number of significant digits in the equations of the calibration curves.
Line 193: Don't separate the value and its SD on two different lines.
Personally I prefer the SD to be expressed in % and as relative SD, RSD.
The Sochaczew treatment plant should be better described by indicating the size of the various treatments and the equivalent number of inhabitants for which it is planned.
After absorption of the compounds by the plants, they become contaminated with the studied compounds and their metabolites.
What solution do the authors foresee for contaminated plants?
From the perspective of the circular economy in the EU this solution must be planned.
Table A2 - The CAS number must be indicated for each compound.
Author Response
Dear Reviewer,
We would like to thank for valuable comments and suggestions. They will definitely help improve the quality of the manuscript. All of them have been taken into consideration. Please find enclosed point by point answers to these comments:
Reviewer 3:
Lines 68-70 and 89-90 repeats the same information.Answer: We fully agree with the Reviewer, so the same information was removed from lines 89-90 in Section 2.
Lines 139-140: do not put in different lines value and units (50 ºC)Answer: The value and unit are in the same line.
Figure 2: The y scale should start at zero, the inner legend of the graphs should be improved.Answer: In both graphs in Figure 2 y scale was corrected and now is started at zero. The inner legend of the graphs were improved according to the Reviewer suggestion.
Table 2: Verify the number of significant digits in the equations of the calibration curves.Answer: According to the Reviewer’s suggestion the number of the significant digits in the equations was verified and corrected.
Line 193: Don't separate the value and its SD on two different lines.Answer: The value and SD are now in the same lines in this case and in the whole manuscript.
Personally I prefer the SD to be expressed in % and as relative SD, RSD.Answer: We have decided to present the SD values in this style because it is recommended in Vocabulary of Metrology. However, we fully agree with the Reviewer that both forms of the SD presentation are correct. If the expression of SD as relative SD, RSD will be crucial for publication we will modify the SD values into the RSD ones.
The Sochaczew treatment plant should be better described by indicating the size of the various treatments and the equivalent number of inhabitants for which it is planned.Answer: The missing information was added to the Section 3.2.
After absorption of the compounds by the plants, they become contaminated with the studied compounds and their metabolites. What solution do the authors foresee for contaminated plants? From the perspective of the circular economy in the EU this solution must be planned.Answer: We are fully agree with the Reviewer that from the perspective of the circular economy in the EU the solution for contaminated plants must be planned. Fortunately, many scientists are working on this solution. One of the option is production of biochar which could be used - for example - as a sorbent. This process involves pyrolysis of contaminated material (here plants) at very high temperature (up to 700 oC). We have also published paper concerning on biochar titled Valuable polar moieties on cereal-derived biochars, in: Colloids and Surfaces A-Physicochemical and Engineering Aspects, vol. 561, 2019, ss. 275-282, DOI:10.1016/j.colsurfa.2018.11.008.
Table A2 - The CAS number must be indicated for each compound.Answer: Now, the CAS number for each target compound is included in Table A2 in Appendix A.
We hope that our explanations will be satisfactory for you and the revised manuscript could be accepted for publication in Molecules.
Yours sincerely,
Jolanta Kumirska

Reviewer 4 Report
Language needs revision Abstract: This section is a summary of the results and research conclusion not of the experimental methods and techniques! Abstract: the problem discussed is not related to MWWTP it is related to Municipal wastewater and it is not resolved in MWWTP line 15: not the analysis of possibility - it is simply the potential to use the plants used are old but were they applied before? What are the investigated chemicals or residues? there should be an indication at least in the abstract. pharmaceuticals and EDCs are not introduced by wastewater plants they are introduced to the environment through municipal wastewater due to excessive use or misuse. Line 73-75: the last sentence discredits the study before it is presented. the aim should be rearranged as 2, 3, 4, and 1 It is not usual that the experimental section to be placed after the results and discussion, As an analytical method is to be presented, it should answer to the linearity of the method, range, LOD, and LOQ, then spike and recovery first, i.e. validation comes first. The study should have commenced by presenting a survey of what is present in real sewage to give substance to the work the whole text needs to be revised and rearranged in a more logical fashionAuthor Response
Dear Reviewer,
We would like to thank for valuable comments and suggestions. They will definitely help improve the quality of the manuscript. All of them have been taken into consideration. Please find enclosed point by point answers to these comments:
Reviewers 4
Language needs revision.Answer: The manuscript was checked and corrected by a native-speaker.
Abstract: This section is a summary of the results and research conclusion not of the experimental methods and techniques!Answer: We are fully agree with the Reviewer. However, according to the Instructions for Authors:
“The abstract should be a total of about 200 words maximum. The abstract should be a single paragraph and should follow the style of structured abstracts, but without headings: 1) Background: Place the question addressed in a broad context and highlight the purpose of the study; 2) Methods: Describe briefly the main methods or treatments applied. Include any relevant preregistration numbers, and species and strains of any animals used. 3) Results: Summarize the article's main findings; and 4) Conclusion: Indicate the main conclusions or interpretations. The abstract should be an objective representation of the article: it must not contain results which are not presented and substantiated in the main text and should not exaggerate the main conclusions.”
We have tried to fill all presented above requirements and for this reason due to limited number of words we have presented only the most important information about 1) Background, 2) Methods, 3) Results, 4) Conclusion.
Abstract: the problem discussed is not related to MWWTP it is related to Municipal wastewater and it is not resolved in MWWTPAnswer: We are fully agree with the Reviewer. For this reason, the first sentence was modified into: The problem of the presence of pharmaceuticals and endocrine disrupting compounds (EDCs) in the environment is closely related to municipal wastewater and in consequence to municipal wastewater treatment plants (MWWTPs) because wastewater is the main way in which these compounds are transferred to the ecosystem.
line 15: not the analysis of possibility - it is simply the potential to use the plants used are old but were they applied before?Answer: The word “ the possibility” was changed into “the potential”. According literature data (Table A1 in Appendix A), these three species of plants are used for the first time for the removal of pharmaceuticals and endocrine disrupting compounds in municipal sewage treatment plants.
What are the investigated chemicals or residues? there should be an indication at least in the abstract.Answer: This information is now included in the Abstract as “….the removal of 15 selected pharmaceuticals and endocrine disrupting compounds (EDCs) in an MWWTP is presented.”
pharmaceuticals and EDCs are not introduced by wastewater plants they are introduced to the environment through municipal wastewater due to excessive use or misuse.Answer: We are fully agree with the Reviewer. For this reason, the first sentence was modified into: “The problem of the presence of pharmaceuticals and endocrine disrupting compounds (EDCs) in the environment is closely related to municipal wastewater and in consequence to municipal wastewater treatment plants (MWWTPs) because wastewater is the main way in which these compounds are transferred to the ecosystem.”
Line 73-75: the last sentence discredits the study before it is presented.Answer: The sentence “For this reason, it is difficult to assume the influence and mechanism of the removal of EDCs from wastewater by plants, as well as to establish which plants are the best for this purpose.” is strongly associated with the previous sentence: “Moreover, in most cases, the removal efficiency of the target compounds from wastewater is determined based only on determining their concentrations in raw and treated wastewater without the examination of their presence in plant tissues [29].” We would like to point out that in most published papers, uptake by plants is not determined, so the influence and mechanism of the removal of EDCs and pharmaceuticals from wastewater is difficult to assume.
the aim should be rearranged as 2, 3, 4, and 1Answer: The main objectives of this work were as follows: 1) to develop an analytical method for determining selected pharmaceuticals and endocrine disrupting compounds in plant materials; 2) to determine the selected pollutants in raw and treated sewage; 3) to assess the bioaccumulation of the mentioned pollutants in hydroponically cultivated plants; 4) to evaluate which of the tested plant species best meets this purpose. According to the Reviewer the order should be modified into: 2) to determine the selected pollutants in raw and treated sewage; 3) to assess the bioaccumulation of the mentioned pollutants in hydroponically cultivated plants; 4) to evaluate which of the tested plant species best meets this purpose. and 1) to develop an analytical method for determining selected pharmaceuticals and endocrine disrupting compounds in plant materials. We are fully agree that the modification of the order of the aims 1 and 2 could be voluble for readers: first the determination of the selected pollutants in raw and treated sewage is presented, next - the development of an analytical method for determining selected pharmaceuticals and endocrine disrupting compounds in plant materials. Such modification is performed in the revised manuscript. However, in our opinion, it is not possible to assess the bioaccumulation of the mentioned pollutants in hydroponically cultivated plants and to evaluate which of the tested plant species best meets this purpose without before presentation of the development of an analytical method for determining these compounds in plant materials. For this reason, the aims in the revised manuscript are presented in the following order: 2, 1, 3, 4.
It is not usual that the experimental section to be placed after the results and discussion, As an analytical method is to be presented, it should answer to the linearity of the method, range, LOD, and LOQ, then spike and recovery first, i.e. validation comes first.Answer: We fully agree with the Reviewer, that it is not usual that the experimental section is placed after the results and discussion. However, according to Instructions for Authors such construction of manuscript is required by Molecules journal.
The study should have commenced by presenting a survey of what is present in real sewage to give substance to the work the whole text needs to be revised and rearranged in a more logical fashionAnswer: As it was mentioned, in the revised manuscript the investigation what is present in real sewage is presented at the beginning of our research, before the development of an analytical method for determining these compounds in plant materials.
We hope that our explanations will be satisfactory for you and the revised manuscript could be accepted for publication in Molecules.
Yours sincerely,
Jolanta Kumirska

Round 2
Reviewer 1 Report
All the suggestions were considered. In my opinion, the tittle can be changed.